# Examination of an Online Cooking Education Program to Improve Shopping Skills, Attitudes toward Cooking, and Cooking Confidence among WIC Participants

**DOI:** 10.3390/nu15194177

**Published:** 2023-09-27

**Authors:** Dena R. Herman, Rachel Kimmel, Skye Shodahl, Jose H. Vargas

**Affiliations:** 1Department of Family and Consumer Sciences, California State University, 18111 Nordhoff Street, Northridge, CA 91330, USA; 2Department of Community Health Sciences, UCLA Fielding School of Public Health, Suite 16-035 CHS, P.O. Box 951772, Los Angeles, CA 90095, USA; 3Department of Psychology, California State University Northridge, 18111 Nordhoff Street, Northridge, CA 91330, USA

**Keywords:** WIC, nutrition education, low-income, program delivery, virtual services, food insecurity, Cooking Matters

## Abstract

The present study examined if adapting the Cooking Matters (CM) curriculum to be used in an online format would improve participants’ shopping skills, attitudes toward cooking, and feelings of cooking confidence, similar to the traditionally offered method, which is conducted in person. Results from factor analyses indicated that the online CM program demonstrated construct and content reliability compared to in-person (Cronbach’s α ≥ 0.70). Repeated-measures ANOVA revealed a decrease in shopping skills overall (F = 5.91; *p* ≤ 0.05), consistent across age groups (F = 3.2; *p* ≤ 0.05) and food security status (F = 7.48; *p* < 0.01), with larger impacts on the food insecure (FI). Positive cooking attitudes increased with income (F = 2.86; *p* ≤ 0.05), especially among the <$20,000 and $30–39,000 income brackets. Cooking confidence increased post-intervention (F = 27.2, *p* < 0.001), with an interaction effect for food security status (F = 7.45; *p* ≤ 0.01), with greater improvement for households with food insecurity. These findings provide evidence to program and policymakers that virtual nutrition and cooking education services for the Special Supplemental Nutrition Program for Women, Infants, and Children (WIC) should continue to be supported beyond the pandemic as they reduce barriers to receiving program benefits, nutrition education, and may lead to reductions in household food insecurity.

## 1. Introduction

The Special Supplemental Nutrition Program for Women, Infants, and Children (WIC) provides supplementary foods, referrals to health care providers, health screenings, breastfeeding support, and nutrition education to low-income mothers and children ages 0–5 [1]. The WIC population is at a high risk of experiencing a lack of access to healthy foods during critical periods of growth and development. This increases their chance of poor health outcomes, including obesity, pregnancy complications, folate deficiency, and gestational diabetes [2]. Higher rates of food insecurity are experienced among households with children under age six (14.3%) and those with incomes below 185% of the poverty level (29.1%), both of which characterize the population eligible to participate in the WIC program [3,4]. Recent data from Los Angeles County WIC reveals that 40% of WIC households were considered food insecure (FI) [5]- defined as inadequate access to enough food to maintain an active, healthy life [6]. This rate is 1.6 times higher than the percentage of food insecurity experienced by the rest of the population in Los Angeles County (24.3% in 2021) [7] and 3.2 times higher than food insecurity rates for households in the United States (10.2% in 2021) [8].

Nutrition education is a unique aspect of the WIC program and is not a required part of any other nutrition assistance program in the United States [9].WIC’s nutrition education is required monthly for WIC recipients to receive benefits. Educational topics include healthy recipes, utilising a limited budget to shop for food, addressing feeding challenges for children, and dietary tips during pregnancy (WIC Basics). In recent years, the WIC program has suffered from lower participation rates, exacerbated during the COVID-19 pandemic [10]. Recent studies show one way to improve WIC participation rates is to offer participants more scheduling flexibility, including the opportunity for hybrid services [11,12].

Teaching kitchens have been shown to provide accessible nutrition education by reaching participants “where they are at” and re-acquainting them with their cultural heritage, foods, and healthful properties [13]. Nutrition education has been offered to WIC participants in actual and virtual formats. It has been found that online nutrition education resulted in equal increases in knowledge, self-efficacy, and healthy behaviors compared to in-person nutrition education delivery [14]. However, online nutrition education had the additional potential benefit of reaching a larger audience and influencing long-term positive dietary changes due to flexibility and ease of access [14,15,16]. WIC’s current nutrition education curriculum does not teach cooking and meal preparation skills, which may be a gap in nutrition education programming.

Cooking Matters (CM) is a national program developed by Share Our Strength, an organization dedicated to ending childhood hunger and poverty in the United States, to provide low-income caregivers with essential skills in shopping for and cooking healthy meals on a budget through hands-on cooking classes, grocery store tours, and digital media [17]. According to social cognitive theory, the face-to-face learning experience in CM’s curriculum leads to improved dietary behaviors, such as more fruit intake and better self-efficacy in terms of food preparation and food resource management [18,19]. Although CM has traditionally been offered in an in-person setting, at the start of the COVID-19 pandemic, greater efforts were devoted to adapting the program to an online format to better support families during these unprecedented times [20]. Moreover, while CM is targeted at low-income families, there are few examples of its use with food assistance program recipients [21] and none that these authors could identify with WIC participants. Therefore, the specific aim of this project was to examine if offering the CM curriculum in an online/virtual format would lead to similar improvements in shopping skills, attitudes towards cooking, and feelings of cooking confidence similar to those demonstrated in the traditionally offered in-person format in a population of WIC participants.

## 2. Materials and Methods

### 2.1. Study Population/Recruitment

Participant recruitment occurred in collaboration with the Ventura County WIC program. Ventura County WIC staff informed their clients of the opportunity to participate in the Family Kitchen cooking education program through monthly mass text and email communications. WIC staff also spoke with eligible clients during regularly scheduled bi-monthly phone visit appointments. It offered the Family Kitchen cooking education program as an option for its clients to meet their nutrition education requirements for the month. Standardized scripts were created in English and Spanish for all recruitment efforts.

Eligibility criteria included being a current Ventura County WIC participant, over 18 and English- or Spanish-speaking. Those interested in participating in the study were directed to the project website (https://familykitchenvc.info, accessed on 5 August 2023), where screening, consent, cooking education videos, and all surveys were located. In consultation with the WIC staff and directorship, it was recommended that the best way to reach clients would be through cell phone access. Thus, all Family Kitchen project content was set up and tested to work equally well on a mobile device and computer. Incentives for Family Kitchen program participation included fulfilling WIC’s nutrition education requirement for the month to receive WIC benefits. Written consent was obtained online and included a section for participants to fill in their name and WIC family ID number if they want to receive credit toward their WIC nutrition education requirement. The present study was approved by the California State University Northridge Institutional Review Board (IRB-FY21-174).

### 2.2. Methods

#### 2.2.1. Intervention Development

The Family Kitchen cooking education videos were created in a pre-recorded format and offered in English and Spanish. The objectives of the cooking education videos included learning how to choose and cook healthy, WIC-approved foods and ways to include children in the kitchen for families living on a budget. The curriculum included five modules prescribed by the funder (Share our Strength/No Kid Left Behind) and an introductory knife skills video. This additional module was developed by Family Kitchen members with the input and guidance of Ventura County WIC staff to ensure study participant safety while cooking in the kitchen with small children.

The topics of the modules included “No More Mealtime Madness”, “Hack Your Snack”, “Drink to Your Health”, “The Family Kitchen”, and “Kids Say Yes to Fruits and Veggies”. Table 1 lists the learning objectives of each module. From these learning modules, seven 20-minute pre-recorded videos in English and Spanish were created and released through the Family Kitchens website once monthly from March 2021 to August 2021. Participants and their children under five were instructed to watch at least one pre-recorded cooking education video and the introductory “Knife Skills” video to meet their WIC nutrition education requirement for the month.

In collaboration with the community partners at Ventura County WIC, recipes were adapted from the CM program to include culturally appropriate, WIC-approved foods for the study population. Given that the population in Ventura County is 44.5% Hispanic or Latino, the recipes that were created included foods traditionally consumed by this study population [22]. Although, in general, the Hispanic and Latino culture emphasizes family rituals, including mealtime rituals, there is no focus on including young children in the kitchen and mealtime preparation [23]. Thus, messages in the pre-recorded cooking education content focused on including children in the kitchen, a beneficial habit for developing healthy eating behaviors for children and the entire family [24]. For each pre-recorded video, standardized scripts were created by the Family Kitchen team using the CM curriculum. English scripts were translated into Spanish and reviewed and back-translated for consistency and understanding by a second Spanish speaker.

#### 2.2.2. Instruments and Measures

The impact of the Family Kitchen program on participants’ healthy eating practices was measured using a prospective, mixed method, pre-post survey design. A screening form was developed to ensure that study inclusion and exclusion criteria were met. The screening, consent, and pre-survey were presented on the project website before participants could access any pre-recorded video content. All surveys and forms were available in English and Spanish and were self-administered online via Qualtrics at the study’s dedicated website (https://familykitchenvc.info/, accessed on 5 August 2023).

To assess changes in shopping skills, attitudes, and confidence in cooking healthy on a budget from pre- to post-intervention, surveys developed from the CM curriculum were used. These surveys included ten validated questions to assess food shopping habits using budget-friendly, nutritious ingredients and attitudes toward and confidence in cooking for the family [25]. Of the ten questions, three assessed shopping skills using a Likert scale, including “never”, “rarely”, “sometimes”, “often”, “always”, “does not apply”. Two questions assessed attitudes toward cooking healthy on a budget using a Likert scale including “strongly disagree”, “disagree”, “neither agree nor disagree”, “agree”, and “strongly agree”. The remaining five questions assessed feelings of confidence in cooking healthy on a budget using a Likert scale of “not very confident”, “neutral”, “somewhat confident”, “very confident”, and “does not apply”.

Additional questions added to the pre-survey to assess the demographic and household characteristics of participants included food security status, sex, education, household income, household size, and race/ethnicity). Food security status was measured using the validated United States Department of Agriculture (USDA) six-item short form Household Food Security Survey Module (Economic Research Service, USDA, 2012). Prevalence of food security was calculated based on the number of affirmative responses to the survey questions, indicating either “yes”, “often”, “sometimes”, “almost every month”, or “some months but not every month”. A score of 0–1 indicated high or marginal food security, 2–4 indicated low food security, and 5–6 indicated very low food security [26].

#### 2.2.3. Data Analysis

Complete pre-and post-surveys in English and Spanish (N = 257) were analyzed using IBM SPSS Statistics software, version 17.0. Criteria for survey completion included missing no more than three survey questions. Frequencies were calculated for age, language, sex, education, income, race/ethnicity, and food security status. To test the first research objective (i.e., to determine the reliability of the study’s measures at both pre-and post-test) in an online/virtual setting compared to the standard in-person cooking education traditionally offered), exploratory factor analysis was conducted on the pre-and post-survey questions. Factor loadings of 0.50 or greater determined which factors on the CM scale grouped (i.e., shopping skills, attitudes, and confidence). Composite scores, an average of the test scores, were calculated for each variable assessed (shopping skills, attitudes, confidence) to determine how well the items on the scale worked together to assess the variable of interest. Cronbach’s α was used to determine survey reliability [27].

To test the second research objective, which was to understand if there was a relationship between participating in a pre-recorded online cooking education program and (1) improved shopping skills to purchase healthy foods on a budget, (2) improved attitudes toward cooking, and/or (3) increased feelings of confidence in cooking habits, change scores were calculated subtracting pre-survey assessment scores from post-survey assessment scores. Data were examined to determine if assumptions for parametric testing were met. Mixed-subjects Analysis of Variance (ANOVA) was used to assess changes from the pre-survey to the post-survey and to identify any relationships between-subjects variables and within-subjects variables. Between-subjects variables included: age, education, race/ethnicity, income, and food security status. The within-subjects variable was the intervention program. Each ANOVA test examined a different dependent variable (i.e., shopping skills, attitudes towards cooking, and confidence in cooking habits). All tests analyzed the main effect of time, the main effect for that particular between-subjects variable, and the interaction effect of time and the between-subjects variable.

## 3. Results

### 3.1. Demographic and Household Characteristics

Table 2 illustrates the demographic and household characteristics of the 257 participants in this study. Among all study participants, 164 individuals (63.8%) were English speakers, and 93 individuals (36.2%) were Spanish speakers, with almost all participants identifying as female (99.2%). Most study participants were 30–39 years of age (51.8%), followed by 30.7% who were 18–29 years old, 15.6% were 40–49 years old, 0.8% were 50–59 years old and 1.2% were 60 years or over. Regarding education, 19.9% completed less than a high school degree, 36.7% completed a high school degree or GED, 22.3% completed some college but did not graduate, 9.6% graduated with a 2-year degree, and 11.6% graduated with a 4-year degree. There were statistically significant differences in years of education completed between English- and Spanish speakers, with Spanish speakers generally completing fewer education than English speakers (Chi-square = 65.6; *p* < 0.001). The racial/ethnic make-up of the sample was predominately Latinx, Mexican, and Hispanic (76.8%), followed by 13% White, 7.4% Asian, 2.8% American Indian or Alaska Native, 1.2% identifying as African-American or Black and the remaining 1.6% as “Other”. There were statistically significant differences between English- and Spanish speakers in terms of their race, with more English speakers identifying as Latinx, Mexican or Hispanic when compared to Spanish speakers (Chi-square = 17.8; *p* = 0.003). In terms of household income, 46.9% earned less than $20,000/year, 22.6% earned between $20,000–$29,999, 14.6% earned between $30,000–$39,999, 8.4% earned $40,000–$49,000, 3.3% earned $50,000–$59,000, and the remaining 4.2% earned $60,000 or more. For 36.7% of study participants, high school was their highest level of education. More than half (55%) of study participants reported experiencing high or marginal food security, 35% reported experiencing low food security, and 9.3% reported experiencing very low food security. There were no other statistically significant differences between English- and Spanish-speakers for the remaining demographic variables.

### 3.2. Factor Analysis

Table 3 shows the results of the exploratory factor analysis from the pre-survey. Varimax rotated factor loadings, a statistical technique used to clarify the relationship among factors reveals four-factor groupings, which are: (1) cooking confidence (Cronbach’s α of 0.80); (2) shopping skills (Cronbach’s α of 0.70); attitudes toward cooking (Cronbach’s α of 0.71); and (3) “nutrition facts label”, which cross-loaded with the other factors. Results of the factor analysis for the post-survey produced varimax rotated factor loadings consistent with those for the pre-survey (results not shown; see Appendix A).

### 3.3. Mixed Subjects ANOVA Analysis

Results from the mixed-subjects ANOVA of the mean composite scores for shopping skills, attitudes toward cooking, cooking confidence, and nutrition facts label reading are shown in Table 4a, Table 4b and Table 4c, respectively. Table 4a shows the results for shopping skills by within-subjects (time) and between–subject factors. There was a decrease in shopping skills from baseline to post-intervention (F = 5.91; *p* ≤ 0.05), which was consistent across age groups (F = 3.2; *p* ≤ 0.05) and food security status (F = 7.48; *p* ≤ 0.01) with a larger decrease among the FI. For the analyses that included education, income, or race, these between-subjects factors did not interact with the within-subjects factor (i.e., time). The estimated marginal means reveal that the 95% confidence intervals overlap for education, income, and race when tested individually against the outcome of interest shopping skills, indicating no differences between these variables.

Table 4b shows results for attitudes towards cooking by within-subjects and between–subjects factors. There was an increase in positive attitudes toward cooking with income (F = 2.85; *p* ≤ 0.05). Specifically, those with incomes less than $20,000/year and those between $30–39,000/year reported the greatest improvements in attitudes. For the analyses that included education, income, or race, these between-subjects factors did not interact with the within-subjects factor (i.e., time). The estimated marginal means reveal that the 95% confidence intervals overlap for the intervention program, age, educational level, and race when tested individually against the outcome of interest attitudes toward cooking, indicating no differences between these variables.

Table 4c shows the results for confidence in cooking skills by within-subjects and between–subjects factors. Results indicate there was an increase in cooking confidence from baseline to post-survey (F = 27.2; *p* < 0.001) regardless of the group variable under analysis and an interaction effect between time and food security status (F = 7.45; *p* ≤ 0.01) with a larger increase in cooking confidence for the FI in comparison to the FS. No relationship was found for age, educational level, income, and race. The estimated marginal means reveal that the 95% confidence interval overlaps for age, income, educational level, and race when tested individually against the outcome of interest confidence in cooking skills, indicating no differences between these variables.

Table 4d shows the results for “knowledge of nutrition facts label use” by within-subjects and between–subject factors. Results show that the use of the nutrition facts label increased on average the most for those 30–39 years old (F = 7.72; *p* ≤ 0.001) and for those who were FI (F = 5.37; *p* < 0.05). There was no interaction between education, income, or race.

## 4. Discussion

Results from this study showed that the CM survey reliably measures the same constructs in an online format as it does in an in-person setting (e.g., with a Cronbach’s α ≥ 0.70, indicating a high degree of reliability). These findings align with a previous study that found high internal consistency of CM surveys in an in-person setting [18]. Reliability analyses also revealed that the “nutrition facts label” question was its construct, even though it is traditionally grouped within the “shopping skills” survey questions. The fact that this item loaded by itself is an indication of the reliability and consistency of the construct, as the CM modules that we were requested to pilot by the funding agency (Share our Strength/No Kid Left Behind) did not include any education information on nutrition facts label knowledge and were therefore not offered as part of our cooking education videos.

The mixed-subjects ANOVA analysis of changes in shopping skills from pre-survey to post-survey showed a statistically significant decrease in scores over time, no matter the age of participants. Additionally, higher-income participants demonstrated a larger decrease in shopping skills for healthy foods on a budget. One explanation for these findings may be that those earning higher incomes compare prices and incorporate budget-friendly meals less often than those living on a limited budget. Additionally, research has found that those with higher incomes spend more on food, especially ready-prepared meals and meals eaten outside of the home at restaurants, indicating that those with increased incomes may have as much of a need for shopping skills as those living on a more limited budget [28]. Results showed that both participants reporting being FS and FI experienced a decrease in shopping skills for healthy foods on a budget from the start of the study to completion. While this result appears counterintuitive, it may be related to the timing of study conduct (March 2021–January 2022). During this period, WIC participants received increased benefits of up to $35 per child and adult per month for fruits and vegetables due to the government’s response to the COVID-19 pandemic, the American Rescue Plan [29]. These increased cash-value vouchers may have reduced the need for participants to shop for budget-friendly foods and/or supported the purchase of what would usually be considered higher priced foods (e.g., fresh fruits and vegetables) as well higher quality protein foods that would have been offset in price due to the additional financial support, which has been shown in other studies [30].

For attitudes toward cooking, participants with the highest income had the greatest increases in scores from pre-test to post-test. These results are consistent with previous research that found those with higher incomes are likely to have more kitchen equipment, more knowledge of how to utilize the cooking equipment and enjoy cooking meals more than those of lower socioeconomic status [31]. In addition, it is essential to note that although income levels are relatively higher, all study participants have incomes that are <185% of the federal poverty level, as this is the income eligibility criteria for WIC participation.

Results also revealed a statistically significant relationship between the intervention program and improvements in confidence in cooking healthy on a budget from pre-survey to post-survey. The food secure (FS) and FI experienced increased confidence in cooking healthy on a budget, with greater increases for those who were FI. These findings are supported by research by Pooler et al. [17], who found that participants receiving benefits from WIC and SNAP experienced a 17% increase in their confidence in managing their budget and purchasing and preparing healthy meals, which was sustained six months after finishing the CM nutrition education program. Similarly, McElwee et al. [32], in their national evaluation of CM programs, found that all participants benefited from the program regardless of their sex, race or educational status, demonstrating that the CM program transcends multiple demographic groups and is an effective program for addressing food insecurity and hunger. While our program did not specifically assess the relationship between food resource management skills as a separate construct and FI, the interaction between cooking healthy on a budget and FI demonstrates the possibility that these constructs are related. Jomma et al. [21] suggest that promoting program participants’ self-confidence in food resource management skills within nutrition education programs such as CM may be explored as a potential strategy to assist low-income households to stretch their food dollars to address household FI.

There are several limitations in this study. First, this study used a correlational design (pre-post design) comparing subjects to themselves from the pre-intervention to post-intervention. While results showed that there was an increase in cooking confidence, improved attitudes toward cooking, and the ability to read nutrition facts labels, it is not possible to discern if these results were due exclusively to the intervention implemented or if there were other reasons for these improvements due to the lack of a control group. Future studies assessing the impact of the CM program should employ a true experimental design to determine if there is a cause-and-effect relationship. However, the original purpose of this study was as a part of a larger national pilot study to understand if offering the CM curriculum in an online environment would be as effective as has been demonstrated in person. As such, the results provided indicate not only the reliability of the main constructs of the CM program but also point to the fact that there may be over time, improvements in these constructs are possible. There are few studies on the CM program and food assistance program recipients [20,32], and none that could be identified by the current authors for the WIC population specifically. Therefore, while this study is not conclusive, it does add to the body of literature that the CM program has the potential to benefit WIC participants and can be reliably administered in an online environment.

Another limitation is that the study population only includes participants from the Ventura County (CA) WIC Program. Although WIC participants have some similar characteristics in terms of demographic information across the nation due to income eligibility criteria, the results of this study may not be generalizable to other WIC programs across the United States. The information in this study may be relevant for other programs that serve high proportions of Spanish speakers or whose program has a large proportion of Latino/a/Hispanic families.

Finally, survey completion necessitated using a mobile device, tablet, or computer with stable internet access. Having internet access may have been a limiting factor for some WIC participants due to the costs associated with and/or accessibility in their region, especially during the COVID-19 pandemic. According to the United States Census Bureau, 94.5% of households in Ventura County have a computer, and 91.1% have a broadband internet subscription [22]. However, Hispanic and Black adults are less likely than those who are white to own a computer or have high-speed internet access [33]. Instead, Hispanics were found to be more likely than any other racial or ethnic group to access the internet solely through their smartphone [33]. Despite this, there is the potential that certain areas of Ventura County, CA, where our participants resided, experienced varying degrees of internet connectivity and cellular reception. In the future, studies of this type should assess internet connectivity and cellular reception to ensure a wide range of WIC families can participate in these studies.

## 5. Conclusions

Over the course of the COVID-19 pandemic, congress allocated 390 million dollars toward enhancing virtual services for WIC sites. There has been strong support among WIC participants for continued virtual services following the pandemic as these have reduced barriers to receiving benefits and nutrition education through WIC and are preferred by program participants [10,12,14,24,34]. The present findings provide support for the benefits of online teaching kitchen models utilizing cooking education curricula such as CM, to not only improve attitudes toward cooking and cooking confidence but also positively affect rates of household FI among low-income and high-risk populations by teaching participants how to purchase, prepare, and serve their families the most nutritious meals they are able on a limited budget [21,30]. While not examined here, research shows that including children in this process from an early age helps to build healthy habits that may serve them into adulthood. Further research assessing the impact of virtual teaching kitchens that deliver cooking and nutrition education for WIC participants is needed to provide evidence to policymakers that virtual services, specifically nutrition education, should continue to be supported.

## Figures and Tables

**Table 1 nutrients-15-04177-t001:** Overview of pre-recorded video content and how many participants watched.

Pre-Recorded Video	Learning Topic	Release Date	Number of Participants that Watched Once (%)
Knife Skills	Tips to buy, use, and care for your knives	21 March 2022	-
No More Mealtime Madness	Planning and preparing quick, chaos-free meals at home (use WIC foods, model how to involve kids in meal preparation)	21 March 2022	33 (12.9)
Hack Your Snack	Encourage making smart choices about snacks (make your snacks focus on fruits and vegetables, model healthy eating habits for kids)	21 April 2022	29 (11.3)
Drink to Your Health	Encourage families to drink more water and less beverages with added sugar	21 May 2022	34 (13.2)
The Family Kitchen	Work together as a family to make healthy meals and snacks (kids are more likely to try new foods when they help choose and prepare them)	21 June 2022	86 (33.5)
Kids Say Yes to Fruits and Veggies	Encourage kids and caregivers to incorporate more fruits and vegetables into family meals and snacks	21 July 2022	42 (16.3)
How to Make Pinto Beans	Essentials of how to cook dried beans	21 August 2022	3 (1.2)

**Table 2 nutrients-15-04177-t002:** Demographic and Household Characteristics of participants (N = 257).

Variable	Total (N = 257) N (%)	English-Speakers (N = 164) N (%)	Spanish-Speakers (N = 93) N (%)	Chi-Square (*p*-Value)
Age (years)				7.03 (0.134)
18–29	79 (30.7)	57 (22.2)	22 (8.6)	
30–39	133 (51.8)	79 (30.7)	54 (21)	
40–49	40 (15.6)	23 (8.9)	17 (6.6)	
50–59	2 (.8)	2 (.8)	0 (0)	
60 and over	3 (1.2)	3 (1.2)	0 (0)	
Gender				2.37 (0.305)
Male	1 (.4)	0 (0)	1 (.4)	
Female	252 (99.2)	163 (64.2)	89 (35)	
Non-binary/third	1 (.4)	1(.4)	0 (0)	
Education Level				65.6 (<0.001 *)
Less than a high school degree	50 (19.9)	15 (6)	35 (13.9)	
High school degree or GED	92 (36.7)	49 (19.5)	43 (17.1)	
Some college, but have not graduated	56 (22.3)	49 (19.5)	7 (2.8)	
Two-year college degree	24 (9.6)	24 (9.6)	0 (0)	
Four-year college degree	29 (11.6)	26 (10.4)	3 (1.2)	
Race				17.8 (0.003 *)
American Indian or Alaska Native	7 (2.8)	6 (2.4)	1 (0.4)	
Asian	12 (7.4)	0 (0)	12 (7.4)	
Black or African-American	3 (1.2)	1 (.4)	2 (0.8)	
White	33 (13)	26 (10.2)	7 (2.8)	
Latinx/Mexican/Hispanic	195 (76.8)	113 (44.5)	82 (32.3)	
Other/Mixed Race	4 (1.6)	4 (1.6)	0 (0)	
Income ($)				3.86 (0.696)
<20,000	112 (46.9)	70 (29.3)	42 (17.6)	
20,000–29,9999	54 (22.6)	33 (13.8)	21 (8.8)	
30,000–39,999	35 (14.6)	25 (10.5)	10 (4.2)	
40,000–49,999	20 (8.4)	14 (5.9)	6 (2.5)	
50,000–59,999	8 (3.3)	7 (2.9)	1 (0.4)	
60,000–69,999	5 (2.1)	4 (1.7)	1 (0.4)	
70,000–79,999	5 (2.1)	3 (1.3)	2 (0.8)	
Food Security Status				0.004 (0.947)
Food Secure	143 (55.6)	91 (35.4)	52 (20.2)	
Food Insecure	114 (44.4)	73 (28.4)	41 (16)	

* Significant at *p* < 0.05.

**Table 3 nutrients-15-04177-t003:** Cooking Matters Pre-Survey: Exploratory Factor Analysis.

Question/Statement	Mean (SD)	Factor Loading 1 (Confidence)	Factor Loading 2 (Shopping Skills)	Factor Loading 3 (Attitudes)
How often do you compare prices before you buy food?	3.19 (0.98)	0.02	0.86	0.08
How often do you adjust meals to include specific ingredients that are more ‘budget friendly’, like on sale or in your refrigerator or pantry?	2.84 (1.03)	0.01	0.85	0.01
How often do you use the ‘nutrition facts’ on food labels?	2.19 (1.17)	0.35	0.43	−0.21
Coking takes too much time.	1.85 (1.08)	−0.10	0.06	0.86
Cooking is frustrating.	1.44 (1.09)	−0.21	−0.05	0.85
How confident are you that you can use basic cooking skills, like cutting fruits and vegetables, measuring ingredients, or following a recipe?	2.48 (0.82)	0.48	0.10	−0.19
How confident are you that you can choose the best-priced form of fruits and vegetables, fresh, frozen, or canned?	2.06 (0.96)	0.73	0.04	−0.09
How confident are you that you can buy healthy foods for your family on a budget?	1.9 (0.98)	0.85	−0.00	0.04
How confident are you that you can cook healthy foods for your family on a budget?	1.86 (1)	0.87	00.04	−0.011
How confident are you that you can help your family eat more healthily?	2.14 (0.93)	0.70	0.08	−0.22
Eigenvalue	-	3.28	1.63	1.29
Cronbach’s α	-	0.80	0.70	0.71

Note: Results of the exploratory factor analysis for the post-survey are similar to the pre-survey. Varimax rotated factor loadings reveal four-factor groupings consistent with the pre-survey. See Appendix A for a table with the results.

**Table 4 nutrients-15-04177-t004:** (**a**) Changes in Shopping Skills from Pre- to Post-Survey by Age, Education, Income, Race/Ethnicity, and Food Security Status. (**b**) Changes in Attitudes Towards Cooking from Pre- to Post-Survey by Age, Education, Income, Race/Ethnicity, and Food Security Status. (**c**) Changes in Confidence in Cooking Skills from Pre- to Post-Survey by Age, Education, Income, Race/Ethnicity, and Food Security Status. (**d**) Changes in Confidence in Nutrition Label Reading from Pre- to Post-Survey by Age, Education, Income, Race/Ethnicity, and Food Security Status.

**(a)**						
**Variable**	**Group**	**Pre-Mean** **(+/− SD)** **THIS IS THE CORRECT SYMBOL TO USE HERE “±”**	**Post-Mean** **(+/− SD)**	**Interaction**	**F**	***p*-Value**
Age	18–29 years	2.86 (0.96)	2.65 (1.05)	Time *	5.91	<0.05
	30–39 years	3.06 (0.89)	0.98 (0.89)	Group	3.2	<0.05
	>40 years	3.12 (0.77)	3.00 (0.85)	Time × Group	0.65	0.52
Educational Level	Less than a HS degree	2.84 (1.1)	2.84 (1.05)	Time *	3.44	0.07
	HS degree or GED	3.00 (0.91)	2.88 (0.95)	Group	0.82	0.44
	Any college	3.11 ± 0.76	2.93 (0.91)	Time × Group	0.84	0.43
Income	<20,000	2.91 (0.92)	2.82 (0.96)			
				Time *	4.56	<0.05
	20,000–29,999	3 (1)	2.87 (1.06)			
				Group	1.21	0.307
	30,000–39,000	3.2 (0.83)	3.09 (0.82)			
				Time × Group	0.08	0.973
	≥40,000	3.14 (0.79)	2.98 (0.86)			
Race/Ethnicity	White	2.93 (0.95)	2.84 (0.99)	Time *	0.4	0.53
	Other	2.79 (0.9)	2.90 (0.89)	Group	0.32	0.73
	Latinx/Hisp/Mex	3.05 (0.89)	2.88 (0.95)	Time × Group	0.25	1.4
Food Security Status	Food Secure	2.86 (0.92)	2.78 (0.98)	Time *	6.41	<0.05
				Group	7.48	<0.01
	Food Insecure	3.19 (0.83)	3.02 (0.89)			
				Time × Group	0.84	0.36
**(b)**						
**Variable**	**Group**	**Pre-Mean** **(+/− SD)**	**Post-Mean** **(+/− SD)**	**Interaction**	**F**	***p*-Value**
Age	18–29 years	1.68 (0.91)	1.62 (0.89)	Time *	0.57	0.45
	30–39 years	1.63 (1.01)	1.48 (0.92)	Group	0.46	0.63
	>40 years	1.63 (0.93)	1.7 (0.83)	Time × Group	1.2	0.3
Educational Level	Less than a HS degree	1.44 (1)	1.4 (0.98)	Time *	1.38	0.24
	HS degree or GED	1.54 (0.82)	1.52 (0.88)	Group	2.64	0.07
	Any college	1.8 (1.02)	1.64 (0.86)	Time × Group	0.72	0.49
Income	<20,000	1.57 (1)	1.5 (0.92)			
				Time *	1.1	0.3
	20,000–29,999	1.44 (1)	1.43 (0.92)			
				Group	2.85	<0.05
	30,000–39,000	1.923(0.8)	1.85 (0.75)			
				Time × Group	0.11	0.95
	≥40,000	1.81 (0.84)	1.7 (0.87)			
Race/Ethnicity	White	1.62 (1)	1.58 (0.94)	Time *	0.68	0.41
	Other	1.6 (0.99)	1.54 (0.73)	Group	0.05	0.95
	Latinx/Hisp/Mex	1.67 (0.96)	1.57 (0.92)	Time × Group	0.95	0.05
Food Security Status	Food Secure	1.59 (0.94)	1.47 (0.9)	Time *	2.24	0.14
				Group	2.56	0.11
	Food Insecure	1.72 (1)	1.68 (0.89)			
				Time × Group	0.38	0.54
**(c)**						
**Variable**	**Group**	**Pre-Mean** **(+/− SD)**	**Post-Mean** **(+/− SD)**	**Interaction**	**F**	***p*-Value**
Age	18–29 years	2.03 (0.71)	2.17 (0.65)	Time *	15.9	<0.01
	30–39 years	2.07 (0.74)	2.3 (0.66)	Group	0.63	0.54
	>40 years	2.15 (0.68)	2.28 (0.65)	Time × Group	0.87	0.42
Educational Level	Less than a HS degree	2.1 (0.81)	2.32 (0.78)	Time *	21.2	<0.01
	HS degree or GED	2.05 (0.68)	2.17 (0.61)	Group	0.5	0.61
	Any college	2.06 (0.71)	2.3 (0.63)	Time × Group	0.87	0.42
Income	<20,000	2.08 (0.75)	2.24 (0.65)			
				Time *	16	<0.001
	20,000–29,999	2.11 (0.76)	2.27 (0.74)			
				Group	0.06	0.98
	30,000–39,000	2.04 (0.67)	2.27 (0.57)			
				Time × Group	0.17	0.92
	≥40,000	2.067(0.65)	2.22 (0.63)			
Race/Ethnicity	White	2.2 (0.67)	2.38 (0.54)	Time *	8.32	<0.05
	Other	2.1 (0.71)	2.18 (0.72)	Group	0.83	0.44
	Latinx/Hisp/Mex	2.04 (0.73)	2.24 (0.67)	Time × Group	0.37	0.69
Food Security Status	Food Secure	2.18 (0.69)	2.27 (0.67)	Time *	27.2	<0.001
				Group	3.11	0.08
	Food Insecure	1.94 (0.74)	2.23 (0.65)			
				Time × Group	7.45	<0.01
**(d)**						
**Variable**	**Group**	**Pre-Mean** **(+/− SD)**	**Post-Mean** **(+/− SD)**	**Interaction**	**F**	***p*-Value**
Age	18–29 years	1.89 (1.16)	1.86 (1.12)	Time *	1.51	0.22
	30–39 years	2.23 (1.22)	2.39 (1.24)	Group	7.72	<0.001
	>40 years	2.6 (0.91)	2.69 (0.92)	Time × Group	1.11	0.33
Educational Level	Less than a HS degree	2.11 (1.2)	2.32 (1.25)	Time *	3.58	0.06
	HS degree or GED	2.33 (1.11)	2.3 (1.14)	Group	0.31	0.74
	Any college	2.12 (1.21)	2.27 (1.21)	Time × Group	1.43	0.24
Income	<20,000	2.15 (1.14)	2.33 (1.17)			
				Time *	0.42	0.52
	20,000–29,999	2.25 (1.28)	2.35 (1.27)			
				Group	1.07	0.37
	30,000–39,000	2.03 (1.2)	1.97 (1.07)			
				Time × Group	1.02	0.38
	≥40,000	2.47 (1.06)	2.42 (1.18)			
Race/Ethnicity	White	2.33 (1.22)	2.42 (1.2)	Time *	0.9	0.34
	Other	2.64 (1.15)	2.64 (1.15)	Group	2.29	0.1
	Latinx/Hisp/Mex	2.11 (1.16)	2.21 (1.18)	Time × Group	0.05	0.96
Food Security Status	Food Secure	2.35 (1.18)	2.43 (1.21)	Time	2.95	0.09
				Group	5.37	<0.05
	Food Insecure	2 (1.14)	2.12 (1.13)			
				Time × Group	0.15	0.7

(a) * “Time” refers to the administration of the assessment survey before (pre) and following (post) the intervention program and is a measure of the within-subject effects (b) * “Time” refers to the administration of the assessment survey before (pre) and following (post) the intervention program and is a measure of the within-subject effects. (c) * “Time” refers to the administration of the assessment survey before (pre) and following (post) the intervention program and is a measure of the within-subject effects. (d) * “Time” refers to the administration of the assessment survey before (pre) and following (post) the intervention program and is a measure of the within-subject effects.

## Data Availability

The data are available upon request.

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
