# Peer review of "Examination of an Online Cooking Education Program to Improve Shopping Skills, Attitudes toward Cooking, and Cooking Confidence among WIC Participants"

_nutrients, 2023, doi:10.3390/nu15194177_

Round 1

Reviewer 1 Report

Please consider the information offered below as an attempt to assist in the refining of your document. In no way are these comments or suggestions meant to challenge the information offered in an aggressive or negative manner. 

Overall, I believe the manuscript is very well written. I found the presentation consistent with high education manuscripts, which made the absorption smooth. I have made some observations that I hope are found helpful or useful. 

Well done. This was a pleasure the review (words not stated often). 

Credentials – 

Spacing issue and font size issue on #4

Abstract – 

Very well done. 

One comment – WIC is used, but not introduced util line 1 of the introduction. 

Introduction  - 

L 47-48 – if you are to offer the percent for LA County, should you not offer the percent for the US as well. 

L 69 – does Share our Strength need qualifying? A citation or brief description. 

L 72 - According to social cognitive theory, the learning experience in CM’s face to face 72 classes leads to improved dietary behaviors, such as more fruit intake, and better self-73 efficacy in terms of food preparation and food resource management [18,19]. 

I question the accuracy of this statement. SCT states nothing about food. Behavior, yes. Food, no. 

Perhaps a rephrasing of this is warranted. 

Materials and Methods 

L 89 – spacing issue before Ventura

Results – 

Very nice. 

Discussion – 

Well done addressing the limitation in the manner you did. Many do not offer as thorough an explanation. That was nice to read. 

Conclusion – 

Very nice. 

L 416 – place a space after 4 and before a 

Reviewer 2 Report

Suggestions

The abstract should begin with the goal of the paper. The sentence: During the COVID-19 pandemic, many food assistance programs pivoted to identify new ways to meet their clients' needs, which is hard to understand. You also mention two methods - one modeled after the in-person Cooking Matters and the traditional in-person method- which needs to be clarified. What is WIC or FI?

Introduction

It is interesting to get information about the benefits of nutritional educational programs, the Special Supplemental Nutrition Program for Women, Infants, and Children (WIC) and the Cooking Matters Program (CM).

Three goals of the paper (line 80) could be a hypothesis. Therefore, rethink the manuscript's general aim in the Abstract and Introduction paragraphs.

Materials and Methods

Why does school not teach children and teenagers how to understand the nutrition rules or use kitchen utensils? We need information on whether children have meals at school and when they are involved in home food preparation (line 132). 

Survey preparation and explanations concerning questions are precious and understandable.

The results paragraph is clear.

Discussion

line 297 text could be more precise.

You should include at least two figures supporting the experiment's concept, results, and discussion of obtained results.

There needs to be more information on other countries working on similar programs and if they have prospects for development.

Literature:

van den Berg, R. SPSS Factor Analysis – Beginners Tutorial Available online: https://www.spss-tutorials.com/spss-factor-analy-491 sis-tutorial/. doesn't exist.

Round 2

Reviewer 2 Report

Small suggestions:

Line 228: I need clarification on that expression. Table 3 shows- it is not the theatre.

The results of …… are presented/are shown in Table 3.

Table 4. a. A dot follows the title.

Table 4. b. The title is without a dot. What is correct?

Line 295: Results from the study showed…. Once more, it is not the theatre. You analyze results to get some idea.

Author Response

Dear Editors,

Please see listed below the response to the requested additional minor edits.  Thank you for your consideration.

Best,

dh

  • Line 228: I need clarification on that expression. Table 3 shows- it is not the theatre.

While I would like to respond to this comment, I am unclear of which expression is being referred to and I do not understand what "it is not the theatre," means.  The word "theatre," does not appear in table 3 or the referenced line 228.

  • The results of …… are presented/are shown in Table 3.

The current statement (line 228) uses the word "shows," not presented.  It is not clear what the reviewer is referring to.  The only use of the word "presented" is on line 144 and refers to our website.

  • Table 4. a. A dot follows the title.

The dot following the title has been removed.  The dot following the number "4" has been removed and only follows the letter "a."

  • Table 4. b. The title is without a dot. What is correct?

There should be no "dot" following the title.  The "dot" following the number "4" has been removed.

  • Line 295: Results from the study showed…. Once more, it is not the theatre. You analyze results to get some idea.

I apologize, but I do not understand what is meant by the word "theatre."  I would be happy to make any revisions necessary as soon as I can understand what is being requested.
